# GATA3 zinc finger 2 mutations reprogram the breast cancer transcriptional network

Motoki Takaku[1], Sara A. Grimm[2], John D. Roberts[1], Kaliopi Chrysovergis[1], Brian D. Bennett[2], Page Myers[3], Lalith Perera [4], Charles J. Tucker[5], Charles M. Perou[6] & Paul A. Wade [1]

GATA3 is frequently mutated in breast cancer; these mutations are widely presumed to be loss-of function despite a dearth of information regarding their effect on disease course or their mechanistic impact on the breast cancer transcriptional network. Here, we address molecular and clinical features associated with GATA3 mutations. A novel classification scheme defines distinct clinical features for patients bearing breast tumors with mutations in the second GATA3 zinc-finger (ZnFn2). An engineered ZnFn2 mutant cell line by CRISPR–Cas9 reveals that mutation of one allele of the GATA3 second zinc finger (ZnFn2) leads to loss of binding and decreased expression at a subset of genes, including Progesterone Receptor. At other loci, associated with epithelial to mesenchymal transition, gain of binding correlates with increased gene expression. These results demonstrate that not all GATA3 mutations are equivalent and that ZnFn2 mutations impact breast cancer through gain and loss-of function.

[1] Epigenetics and Stem Cell Biology Laboratory, National Institute of Environmental Health Sciences, Research Triangle Park, Durham, NC 27709, USA. [2] Integrative Bioinformatics, National Institute of Environmental Health Sciences, Research Triangle Park, Durham, NC 27709, USA. [3] Comparative Medicine Branch, National Institute of Environmental Health Sciences, Research Triangle Park, 27709 Durham, NC, USA. [4] Laboratory of Genome Integrity and Structural Biology, National Institute of Environmental Health Sciences, Research Triangle Park, Durham, NC 27709, USA. [5] Fluorescence Microscopy and Imaging Center, National Institute of Environmental Health Sciences, Research Triangle Park, Durham, NC 27709, USA. [6] Lineberger Comprehensive Cancer Center and Department of Genetics, University of North Carolina, Chapel Hill, NC 27599, USA. Correspondence and requests for materials should be addressed to P.A.W. (email: wadep2@niehs.nih.gov)

Breast cancer is an important cause of cancer mortality among women. Transcriptomic data classifies breast cancer into six subtypes—(1) Luminal A; (2) Luminal B; (3) HER2 positive; (4) Basal-like; (5) Claudin-low; and (6) Normal breast-like—that differ not only in molecular characteristics but also in disease course and response to therapy[1–3]. Systems-level analyses have identified GATA3 as one of the most frequently mutated genes in breast cancers[4,5], yet the function of GATA3 mutations in breast tumors is poorly understood.

GATA3 belongs to the zinc-finger transcription factor family that functions as a key regulator of multiple developmental pathways including mammary epithelial cell differentiation[6–10]. In breast cancer, the expression level of GATA3 is strongly associated with estrogen receptor alpha (ERα)[11,12], and loss of GATA3 expression is associated with poor prognosis[13,14]. In both animal and human cell line models, GATA3 functions as a tumor suppressor by inducing epithelial and suppressing mesenchymal fates[15–17]. GATA3 acts as a pioneer transcription factor during mesenchymal-to-epithelial transition[18]; chromatin binding of GATA3 is important for the recruitment of other co-factors such as ERα and FOXA1 in breast cancer cells[19,20].

Based on the The Cancer Genome Atlas (TCGA) data cohort, approximately 10% of breast tumors harbor somatic mutations in the GATA3 gene[5,21]. These mutations are typically heterozygous and highly concentrated in the C-terminal region of GATA3, where the DNA-binding domain is located. The high frequency suggests that GATA3 mutations are cancer "drivers". Mutations in the second zinc finger domain cause alterations of DNA-binding activity and protein stability of GATA3[22–24]. However, it is still largely unknown how GATA3 mutations influence broader breast cancer properties such as changes in gene regulatory networks and tumor growth[25].

Here we examine the impact of GATA3 mutations on disease course by establishing a novel classification strategy. We find that one specific class of mutation, frame-shift mutations in the second zinc finger, lead to poor outcome when compared to GATA3 wild type or other classes of GATA3-mutant tumors. Utilizing genome editing, we develop a model to study the molecular outcomes of frame-shift mutations in the second zinc finger of GATA3 in breast cancer. The R330 frame-shift mutation leads to alterations in cell morphology consistent with a partial epithelial to mesenchymal transition and to a growth advantage in a xenograft model. At the molecular level, mutation of one allele of GATA3 induces redistribution of GATA3 at roughly 25% of its genomic sites of accumulation. Loci gaining GATA3 occupancy in the mutant cells tend to have increased expression and correlate with genes integral to epithelial to mesenchymal transition. Loci losing GATA3 occupancy tend to have decreases in expression, to associate with epithelial phenotypes and include the progesterone receptor. Accordingly, GATA3-mutant cells have a blunted response to the growth arrest induced by progesterone and exhibit abnormal regulation of a substantial subset of the progesterone-responsive transcriptome. These results shed new light on the impact of GATA3 mutations on breast cancer at the cellular and molecular levels.

## Results

### Distinct features of GATA3 ZnFn2 mutations. 
In breast cancer, GATA3 expression is a prominent marker of luminal breast tumors, and loss of GATA3 expression is associated with aggressive tumor phenotypes. Utilizing the gene expression data from the largest available breast cancer data cohort: the Molecular Taxonomy of Breast Cancer International Consortium (METABRIC)[4], we created two patient groups based on GATA3 gene expression (Fig. 1a). Consistent with the previous literature,

breast tumors with lower GATA3 expression showed significantly worse prognosis than tumors with higher GATA3 expression (Fig. 1a). Within high GATA3 expression cases, patients harboring GATA3 mutations represent better prognosis than GATA3 wild-type cases (Fig. 1a), suggesting that GATA3 mutations are not simple loss-of-function mutations.

GATA3 mutations are localized predominantly in Exons 5 and 6, impacting coding regions and splice sites (Fig. 1b; Supplementary Fig. 1a–c). The majority are insertion/deletion (indel) mutations that induce frame-shifts or alternative splicing, resulting in protein truncation or extension (Supplementary Fig. 1d). Based on the predicted protein products, we classified GATA3 mutations into five groups: (1) ZnFn2 mutations, (2) splice site mutations, (3) truncation mutations, (4) extension mutations, and (5) missense mutations (Fig. 1b; Supplementary Fig. 1d). To assess features within each mutant group, we analyzed the distribution of breast cancer intrinsic subtypes in the METABRIC cohort. As expected, tumors with high GATA3 expression were frequently observed in luminal tumors, while low GATA3 expression cases were often observed in the basal subtype (Fig. 1c). Among 1980 patient cases, 230 tumors harbored GATA3 mutations, and 75% of the mutations were observed in luminal tumors (47% in luminal A, 28% in luminal B). GATA3 mutations were very infrequent in basal tumors (Fig. 1c). Interestingly, ZnFn2 mutations were distinct amongst GATA3 mutations, as they were predominantly found in luminal B tumors (52%, 16 out of 29 cases) (Fig. 1d; Supplementary Fig. 1e), whereas splice site and truncation mutations were frequently observed (62% or 68%, respectively; Supplementary Fig. 1e) in luminal A tumors. The distributions of extension and missense mutations were similar to that of GATA3 wild type. To confirm these observations in an independent cohort, we explored the TCGA data set. Again, tumors bearing ZnFn2 mutations were typically categorized as luminal B, and the splice site mutant tumors were often categorized as luminal A (Supplementary Fig. 1e,f). Because luminal B tumors tend to have worse prognosis than luminal A[3], we performed survival analysis from records in the METABRIC data. Within high GATA3 expression cases, patients carrying ZnFn2 mutations had significantly worse survival (10-year survival rate) than patients with other GATA3 mutations (Fig. 1e). The differential clinical outcomes between ZnFn2 mutations vs. splice site mutations indicate that a mutation in the ZnFn2 motif may be more damaging than complete loss of this domain (Supplementary Fig. 1g).

### ZnFn2 mutation induces transcriptional reprogramming. 
To dissect the impact of GATA3 mutations in breast cancer properties, we decided to focus on ZnFn2 mutations, because (1) gene expression profiles of established luminal breast cancer cell lines are known to be classified into the luminal B subtype[26], and the ZnFn2 mutation group is the only group that is significantly frequently classified as luminal B; (2) the ZnFn2 mutations are associated with poor clinical outcomes. The frame-shift mutation at arginine 330 (R330fs) is the only mutation found in all four breast cancer data cohorts examined (Fig. 1b; Supplementary Fig. 1a–c). The mutations at R330 are typically heterozygous (9 out of 10 cases) and frequently are 1-nucleotide insertions or 2-nucleotide deletions, resulting in a premature stop codon at the same chromosome position (Chromosome 10: 8,111,563-8,111,565). A model structure of the DNA-binding domain of the R330fs mutant protein suggested altered DNA-binding properties of the mutant due to a significant conformational change near the Zn-binding region of the second zinc finger (Supplementary Fig. 2a). To model the situation observed in patients, we introduced a 2-nucleotide deletion at R330 into T47D breast cancer

cells using CRISPR–Cas9 (Fig. 2a,b; Supplementary Fig. 2b,c)[27]. The mutant cell clone (CR3) expressed both wild type and mutant GATA3 proteins endogenously (Fig. 2c).

The proliferation rate of mutant cells was comparable to control T47D cells in vitro (Supplementary Fig. 2d), while the mutant cells exhibited different morphology as compared to control cells (Fig. 2d). The mutant (CR3) cells formed less rounded colonies, and exhibited increased cell spreading and edge protrusion as well as distinct cell-to-cell boundaries. To measure the impact of this mutation on tumor growth in vivo, we conducted xenograft experiments. T47D cells bearing either wild-type GATA3 or the R330fs mutation were injected into the

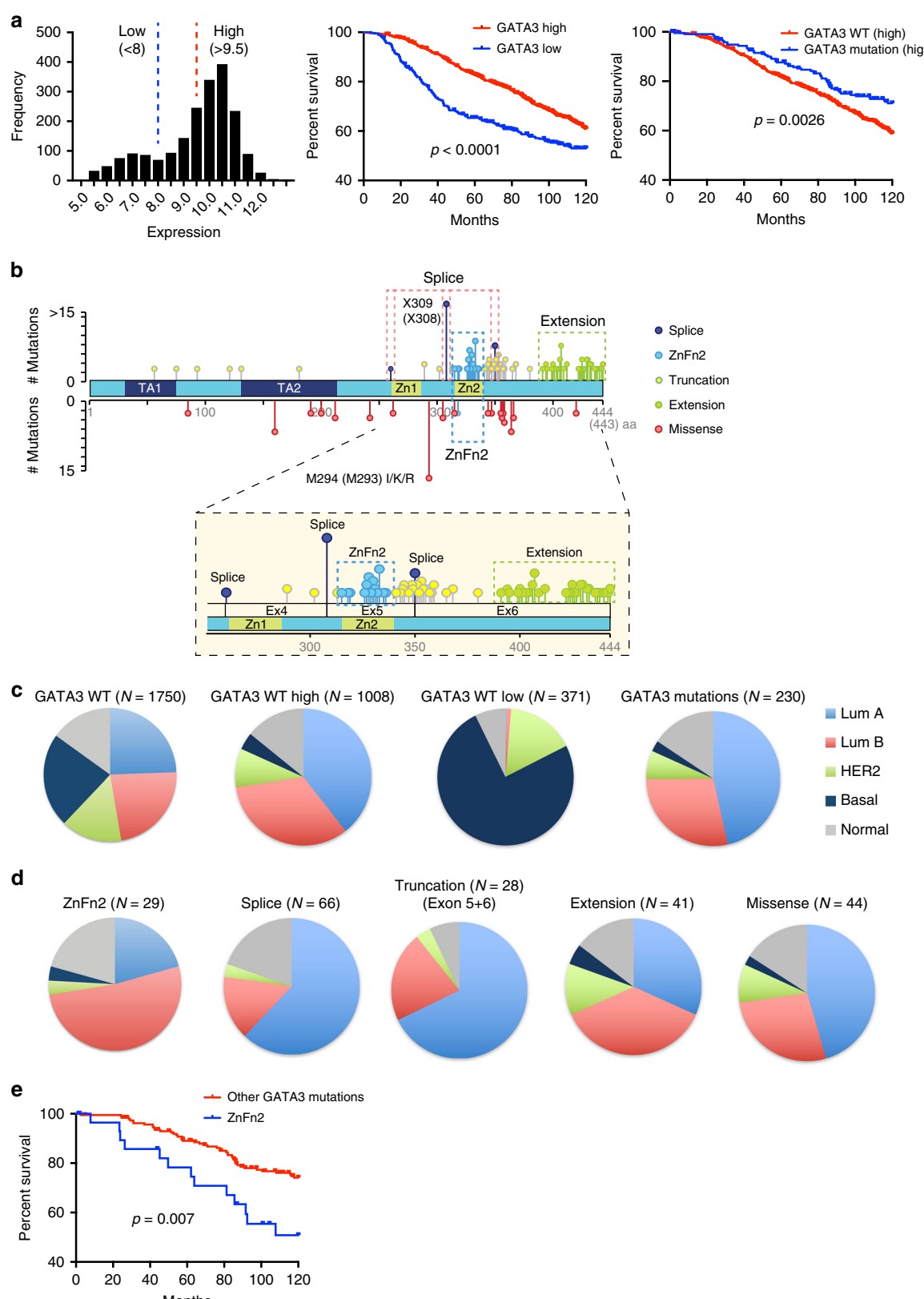

mammary fat pads of nude mice carrying subcutaneous estrogen pellets. After 3 to 5 weeks, GATA3-mutant tumors exhibited significantly higher luminescent signals than controls, indicating that the R330fs mutant tumors had a higher growth rate in vivo (Fig. 2e).

To investigate this phenotypic alteration at a molecular level, we conducted RNA-seq analysis from in vitro cultured cells. At a false discovery rate (FDR) < 0.01, |fold change| > 2, we identified 1173 differentially expressed genes (DEGs, 601 genes upregulated, 572 genes downregulated) (Fig. 3a; Supplementary Data 1). Clustering analysis showed distinctly different gene expression patterns between wild-type and GATA3-mutant cells (Fig. 3b). We performed gene pathway analyses on the DEGs to extract upregulated and downregulated functional networks in the mutant cells. Upregulated genes, including the epithelial to mesenchymal regulators TWIST1 and SNAI2 (SLUG)[28], are associated with cell movement-related and cell invasion-related pathways, consistent with the phenotypes observed in the mutant cells (Fig. 3c; Supplementary Fig. 2e). On the other hand, downregulated genes, including progesterone receptor (PGR), are enriched with cell differentiation-related and development-related pathways, suggesting altered properties in the mutant cells (Supplementary Fig. 2e,f). Analysis of upstream regulators of DEGs indicated downregulation of the progesterone receptor pathway (Fig. 3d), suggesting that the decreased PGR mRNA levels observed in the GATA3-mutant cells has a functional outcome. We also performed RNA-seq analysis of tumors excised from the mouse xenograft model. Although the global gene expression profile was distinct from that derived in vitro, the expression patterns within the DEGs identified by in vitro cell system were largely conserved in the xenografts (Fig. 3e,f; Supplementary Fig. 2g). Finally, gene ontology analyses with cancer data panels revealed common gene signatures between the mutant cells and aggressive tumors (Supplementary Table 1, 2). For example, from the comparison between differentially down-regulated genes (572 genes) in GATA3-mutant cells and the "Ductal Breast Carcinoma Epithelia - Dead at 1 Year[29]" clinical gene expression data, the expression data from 351 genes were available, and 84 genes were significantly ($p < 0.05$, Welch's $t$-test) downregulated in the patients deceased at 1 year. Among 84 genes, 69 genes were categorized in the top 5% under-expressed genes (Fig. 3g; Supplementary Table 2). Taken together, the systematic gene expression data analyses indicated that more aggressive gene signatures were found in GATA3 ZnFn2 mutant cells.

To address potential off-target effects of the CRISPR–Cas9 system, we established a stable cell pool expressing the R330fs mutant exogenously (ExoR330fs) (Supplementary Fig. 3a). The principal features observed in the CRISPR clone (CR3) were also detected in the ExoR330fs cells (Supplementary Fig. 3b–d). In addition, we generated a control clone (CRctrl) to rule out any potential off-target effects (Supplementary Fig. 3a) by using the wild-type GATA3 gene template as the donor DNA. Analysis of select candidate DEGs suggest that the contributions of clonal specificity and off-target effects to the cellular phenotypes were not significant (Supplementary Figure 3e).

To further confirm the impact of the R330fs mutant, we exogenously expressed the mutant in two additional luminal breast cancer cell lines, HCC1428 and BT474. In both cell lines, the R330fs mutant expression led to significant reduction of PR as well as other hormone receptor related genes (Supplementary Fig. 3f–g). Collectively, our in vitro model system suggested that the transcriptome reprogramming induced by the ZnFn2 mutation had substantial impacts on breast cancer properties including tumor growth.

**R330fs mutation induces relocalization of GATA3**. The fundamental difference between wild-type and mutant cells is a 2-nucleotide deletion at the second zinc-finger domain of GATA3. This deletion was predicted to alter DNA-binding activity (Supplementary Fig. 2a) and, thus, we hypothesized, to alter the genomic distribution of GATA3. To address this question, we performed ChIP-seq analyses using an N-terminal specific GATA3 antibody, which recognizes both wild-type and mutant proteins. We identified approximately equal numbers of GATA3-binding sites in wild-type and mutant cells. To dissect differential binding, we classified the peaks into 3 groups: (1) increased: increased ChIP-seq signals in the mutant cells, (2) decreased: decreased signals in the mutant cells, and (3) unchanged: conserved signals between the cells. At an FDR < 0.05 and |fold change| > 1.5, we observed 5161 increased peaks and 5954 decreased peaks (Fig. 4a–c). Most binding sites (~75%) are unchanged between the wild-type and mutant cells.

To evaluate the impact of differential-binding events on gene expression, we assigned the peaks to the nearest transcription start sites (TSS) and performed Gene Set Enrichment Analysis. Increased peaks were strongly associated with upregulated genes including TWIST1 (Fig. 4a,d; Supplementary Fig. 4b). Decreased peaks were associated with downregulated genes including PGR (Fig. 4a,d; Supplementary Fig. 4a,b). This bias in transcriptional outcome was not observed in genes associated with unchanged peaks.

To elucidate the impact on chromatin structure, we performed ATAC-seq[30]. Scatterplot and metaplot analyses of each group revealed a positive correlation between GATA3-binding and chromatin accessibility (Fig. 4e; Supplementary Fig. 4c,d), consistent with its function as a pioneer factor[18,19]. The signal levels at baseline in the decreased peak group were lower than those in the increased peak group (Supplementary Fig. 4e), suggesting a potential difference in chromatin architecture.

We further investigated sequence enrichment at the differential-binding sites. GATA3 is known to recognize the WGATAR motif; therefore, we explored the frequency of this consensus motif in each binding group (Supplementary Fig. 4f). Interestingly, the increased-binding peaks contained less of the consensus motif than did the decreased binding peaks, implying the R330fs mutant might have altered DNA-binding properties. These results suggest that relocalization of GATA3 induced by the

---

**Fig. 1** ZnFn2 mutant tumors are frequently observed in luminal B breast cancers and have worse survival. **a** GATA3 mutations associate with favorable prognosis. Histogram shows distribution of GATA3 expression in the METABRIC cohort (left). Patients are categorized into low (<8) or high (>9.5) GATA3 expression group, and these groups were applied for Kaplan–Meier survival analyses (10-year survival). High GATA3 expression cases were used for the survival analysis shown in right panel. Log rank $p$-values are indicated in the panels. **b** Distribution of GATA3 mutations found in breast tumors. The mutation data was obtained from the METABRIC cohort via the cBio Cancer Genomics Portal. The positions of zinc-finger motifs were obtained from the Uniprot database[42]. **c,d** Subtype frequency in each GATA3 expression/mutation class. The PAM50 classified subtypes were used to generate pie charts. **e** Kaplan–Meier 10-year survival analyses of patients expressing GATA3 mutations. "Other GATA3 mutations" indicates the cases that possess any GATA3 mutations other than ZnFn2 mutations. Log rank $p$-values are indicated in the panels. Higher GATA3 expression cases were used to generate the survival curve

R330fs mutation contributes to the global transcriptional alterations in breast cancer cells and, importantly, that the impact of the relocalization is due to a subset of the GATA3 sites. Determining the molecular rationale for why specific GATA3 sites are critical to phenotype and outcome thus becomes an important goal for deciphering the role of GATA3 in cancer.

**Genomic distribution of mutant GATA3 differs from wild type**. Although ZnFn2 mutations were predicted to have loss-of-function, our ChIP-seq data suggested both gain-of-function and loss-of-function at specific genomic loci. To further investigate, we determined the localization of the mutant in ExOR330fs cells by using the Ty1 epitope tag antibody (Fig. 5a; Supplementary Fig. 5a), identifying 6239 peaks. Motif enrichment analysis revealed a unique-binding motif "AGATBD" (Fig. 5b). R330fs-binding signals were higher in the increased total GATA3-binding group as compared to the other groups (Fig. 5c). We also performed wild-type GATA3 specific ChIP-seq using an antibody, which binds to the C-terminal region of GATA3, thereby recognizing wild-type GATA3 but not the R330fs mutant (Fig. 5a). As expected, the wild-type-specific binding peaks contained a significant enrichment of the consensus-binding motif both in wild-type and mutant cells (Fig. 5b). Peak overlap analysis indicated co-localization of both wild-type and mutant GATA3 at 2944 loci, the unique presence of the R330fs mutant alone at 3295 loci, and wild-type GATA3 unique localization at 6751 loci

(Fig. 5d,e). The "AGATBD" motif was significantly enriched in R330fs unique peaks (Fig. 5f; Supplementary Fig. 5b) while the consensus motif was depleted. To confirm the mutant-specific localization in the endogenous expression cell system, we analyzed the total GATA3 ChIP-seq data along with R330fs and wild-type GATA3 ChIP-seq data. At a subset of R330fs unique peak regions, we observed increased binding signals from total GATA3 ChIP-seq but not from wild-type GATA3 ChIP-seq in mutant cells, suggesting unique localization of the mutant when expressed under physiological cell system (Supplementary Fig. 5a). Consistently, metaplot analysis at R330fs unique peaks showed increased levels of total GATA3 binding in mutant cells compared to wild-type cells, while the wild-type GATA3 binding was decreased at those peaks (Supplementary Fig. 5c). These data indicated that the endogenously expressed R330fs mutant had a similar distribution pattern to the exogenously expressed protein. Metaplot analysis of wild-type GATA3 ChIP-seq data at different GATA3-peak classes also suggested altered distribution of wild-type GATA3 in mutant cells (Supplementary Fig. 5d). Taken together, the genomic profiling suggested novel GATA3-binding events induced by the R330fs mutant. Importantly, both RNA-seq and ChIP-seq data indicated that those gained GATA3-binding events led to activation of cell migration-related and EMT-related pathways. Consistent with this data, mutant cells exhibit an altered phenotype more consistent with mesenchymal characteristics, including cell shape, cell boundaries, and size/distribution of actin stress fibers (Figs. 2d and 5g,h).

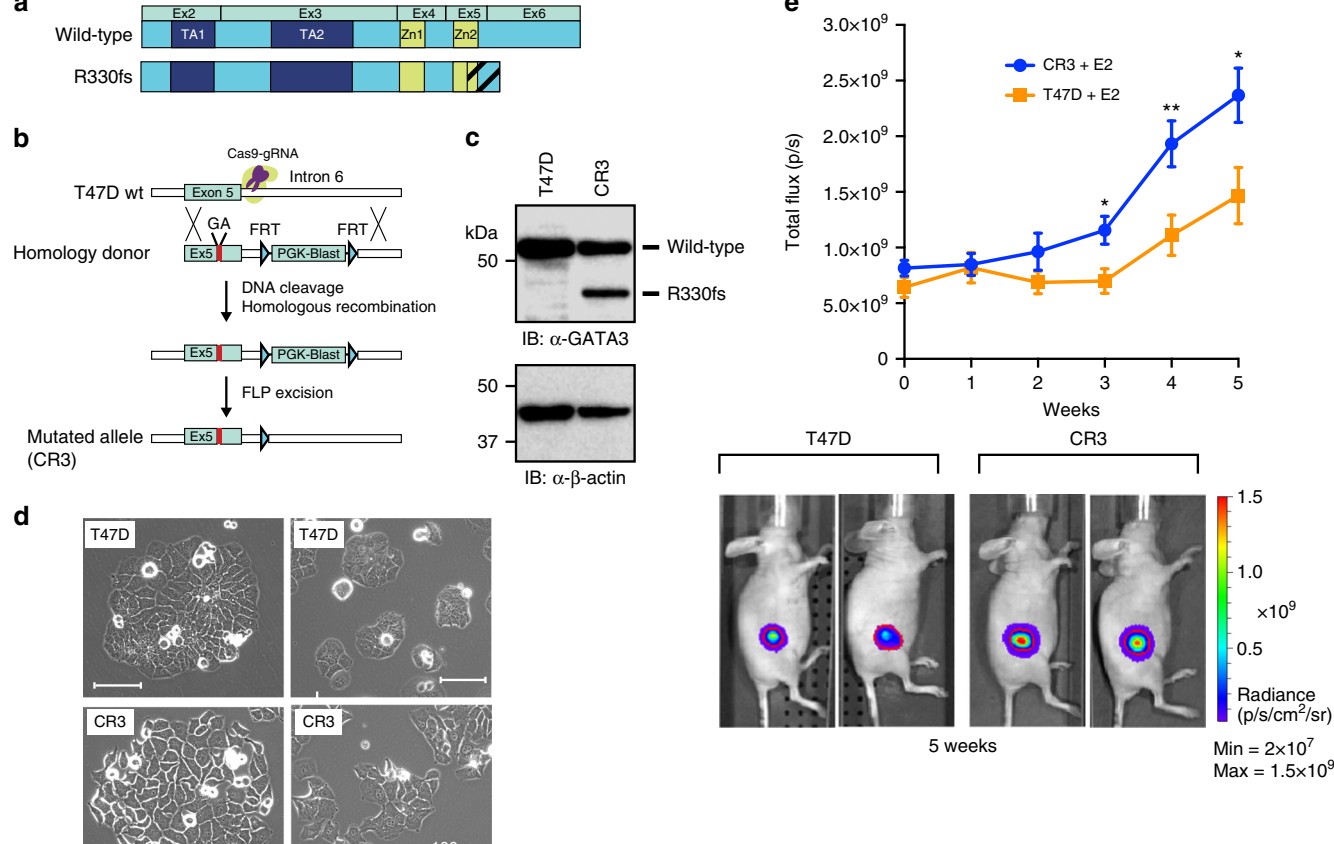

**Fig. 2** GATA3 R330fs mutation facilitates tumor proliferation. **a** The protein domain structures of wild-type GATA3 and R330fs mutant are indicated. **b** Schematic diagram of genome editing by the CRISPR–Cas9 method. **c** Immunoblot analysis of the R330fs mutant clone (CR3). **d** Representative images of control T47D and CR3 cells. The scale bar indicates 100 μm. **e** Tumor growth in the mouse xenograft model. The average bioluminescence signals are quantified and plotted with SEM as error bars ($N > 12$). E2: estradiol pellet. Representative images are shown at the bottom. The red outline over tumors encompasses 97% of total bioluminescent signal in each animal. *$p < 0.05$, **$p < 0.01$, unpaired $t$-test

**ZnFn2 mutations attenuate progesterone receptor signaling.** Next, we sought to investigate molecular and phenotypic outcomes induced by loss of GATA3 binding. One of the most striking losses of GATA3 binding was observed upstream of the PGR locus (Fig. 4a). Consistently, PR gene expression was downregulated in the mutant cells (Supplementary Fig. 3c). PR expression is a critical prognostic marker in breast cancer, and lower expression of PR is associated with poor prognosis[31]. We confirmed that PR expression was dramatically reduced at the protein level in the mutant T47D cells (Fig. 6a). The luminal breast cancer cell lines, MCF7 and KPL-1, are known to possess recurrent ZnFn2 mutations (D336fs)[24,32,33]. We confirmed these GATA3 frame-shift mutations were present in our MCF7 and KPL-1 cells (Supplementary Fig. 6a). The expression levels of PR in these cells were also significantly lower than those in T47D cells (Supplementary Fig. 6a).

We then evaluated potential enhancer function of the GATA3-bound region upstream of PGR. The CRISPR–Cas9 technique was utilized to delete specific chromatin regions. The gRNAs were designed to target two major peaks (Pk1, Pk2) and a control region (Ctrl) (Fig. 6b; Supplementary Fig. 6b). PR expression in the Pk1-depleted cells was reduced, but the Pk2-depleted cells and control cells did not exhibit a significant reduction (Fig. 6b; Supplementary Fig. 6c). These results implicate the Pk1 locus as rate-limiting for PR expression.

Progesterone, a ligand for progesterone receptor, is reported to have anti-proliferative function in PR-positive breast cancers in vivo and in vitro[34–36]. Given our results with the proposed PR-enhancer deletion, we treated the T47D cells with progesterone and measured cell proliferation. Consistent with previous reports, growth of control T47D cells was strongly inhibited by progesterone (Fig. 6c,d). However, this anti-proliferative effect was significantly weaker in the R330fs mutant cells (Fig. 6c,d). A similar weak inhibition was also detected in MCF7 cells, which also bear a ZnFn2 mutation. To confirm the importance of PR action in the mutant cells, we transduced CR3 cells with PR-A

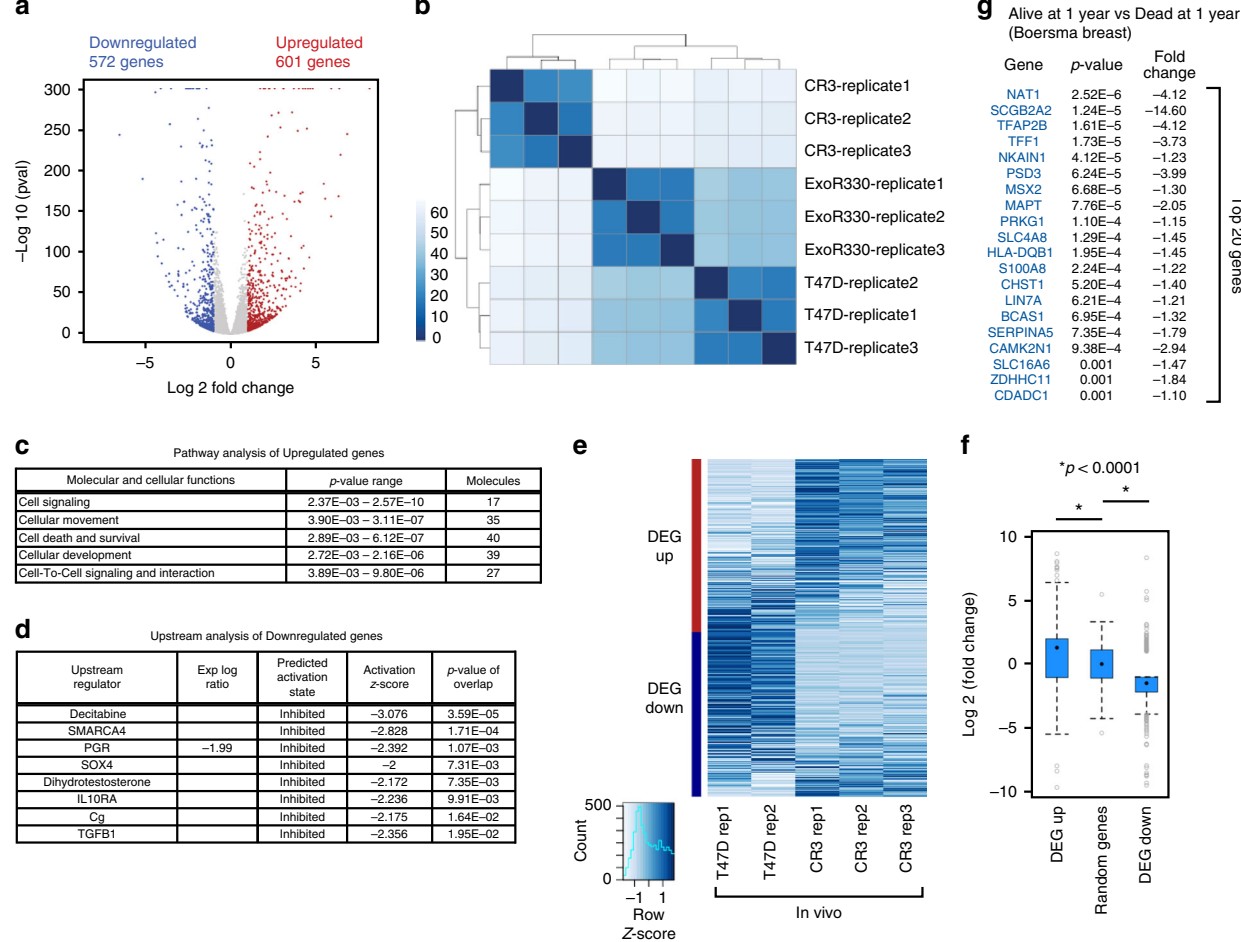

**Fig. 3** R330fs mutation reprograms the luminal transcriptional program in breast cancer cells. **a** Volcano plot displaying the differentially expressed genes between control and GATA3-mutant (CR3) cells. Upregulated genes in CR3 cells are highlighted in red, and downregulated genes are highlighted in blue. Genes with -log$_{10}$ p-values > 300 are all plotted at 300 on the ordinate. **b** Clustered heatmap showing expression patterns in T47D, CR3 and ExoR330fs cells from three biological replicates. The scale indicates Euclidean distance. **c** The top molecular and cellular function related pathways. The IPA (Ingenuity Pathway Analysis) analysis was conducted by using top 100 upregulated genes (ranked by fold changes), and the top 5 pathway categories are presented. **d** The upstream regulators predicted by the IPA (QIAGEN) of downregulated genes are listed. Progesterone receptor was identified as one of the downregulated factors in CR3 cells. **e** Heatmap showing the row-scaled FPKM values of the in vivo RNA-seq data. The differentially expressed genes identified by the in vitro RNA-seq shown in **a** are categorized into upregulated and downregulated gene groups. **f** Box-and-whisker plots showing relative fold changes in each category. Random genes indicate the randomly selected subset of 600 genes that were not categorized as upregulated or downregulated genes. **g** Top 20 genes identified by Oncomine concept analysis. Downregulated genes in CR3 cells were used as input gene set for the gene ontology analysis (Oncomine). 84 genes were significantly downregulated in the tumors from the patients who died at 1 year. Genes are ranked by p-values

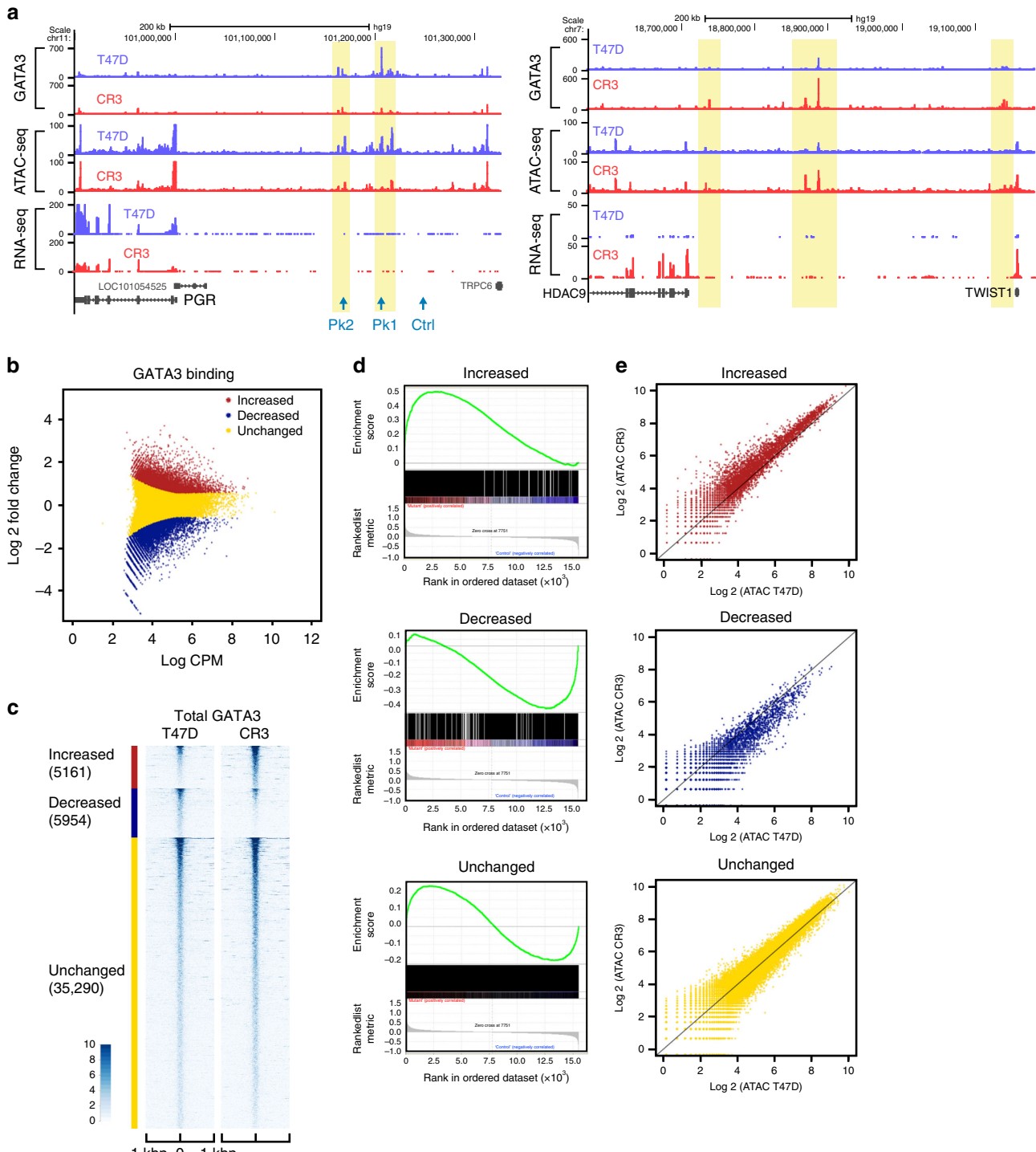

**Fig. 4** R330fs GATA3 mutation induces genome-wide relocalization of GATA3. **a** Representative genome browser tracks at differential GATA3-binding loci. UCSC Genome Browser views showing the mapped read coverage of ChIP-seq, ATAC-seq, and RNA-seq data. The differential-binding sites are highlighted in yellow. The read densities in T47D and CR3 cells are presented in blue and red, respectively. **b** Differential-binding analysis of GATA3-binding events in control T47D versus CR3 cells. Increased and decreased binding events in CR3 cells are indicated in red and blue, respectively. The conserved-binding events are presented in yellow. CPM indicates counts per million in each peak region. **c** Read density heatmaps showing the signal intensities of GATA3 ChIP-seq in T47D (left) or CR3 (right) cells. The number of peaks in each category is reported below the group label. Each row indicates a 2 kbp window centered on a GATA3-binding site. The scale of read density after normalization is indicated at the bottom left. **d** GSEA analysis showing the correlation between each class of GATA3-binding and gene expression. **e** Scatter plots of normalized ATAC-seq signals in each GATA3-peak group. Normalized read counts in CR3 cells (Y-axis) were plotted against the counts in T47D cells (X-axis)

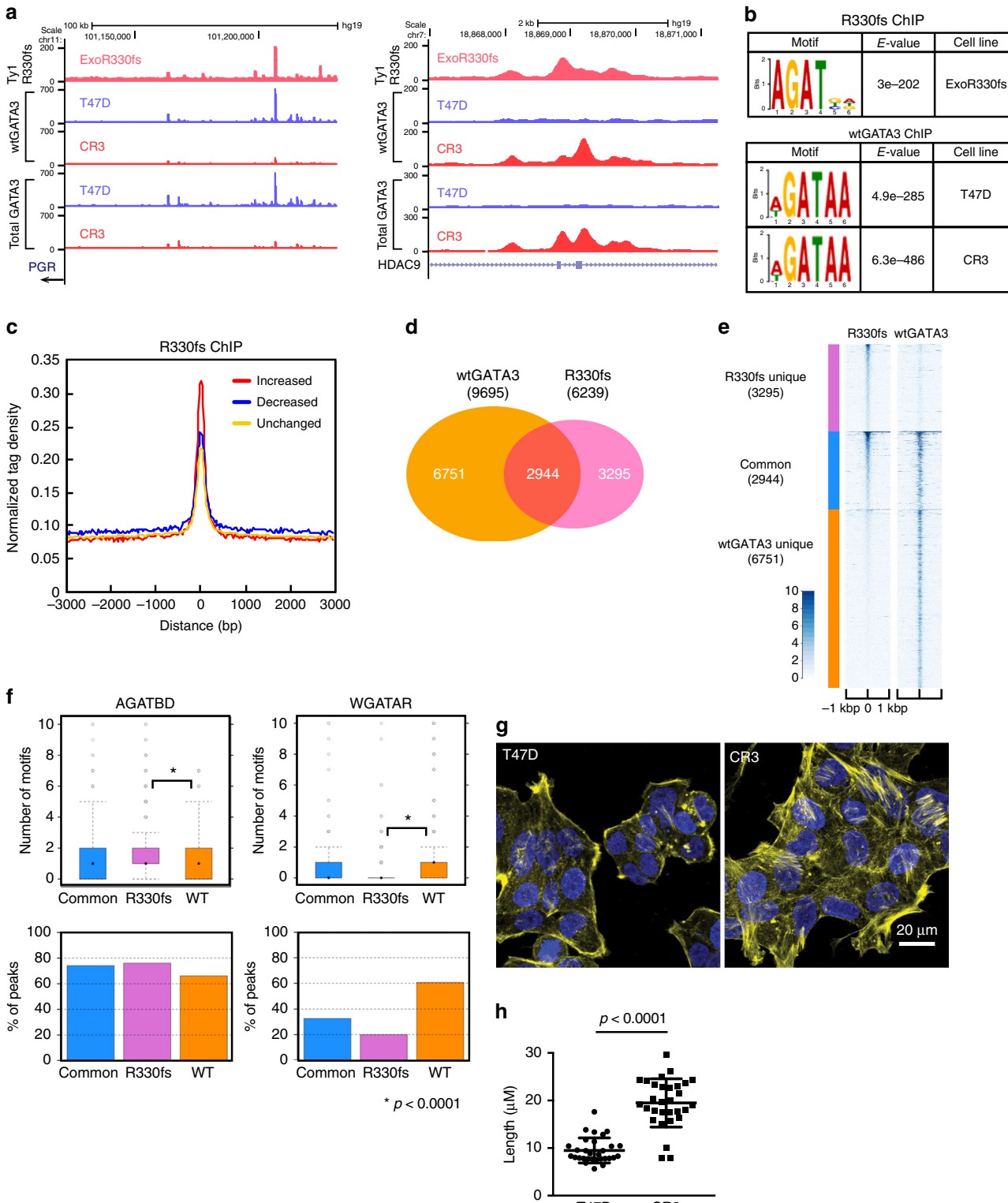

**Fig. 5** Aberrant chromatin-binding property of R330fs mutant influences GATA3 localization. **a** Representative genome browser tracks at R330fs-mutant-binding loci. UCSC Genome Browser views showing the mapped read coverage of R330fs mutant, wild-type GATA3, and total GATA3 ChIP-seq data. The read densities in T47D and CR3 cells are presented in blue and red, respectively. **b** Top-ranked motif at R330fs and wild-type GATA3-binding sites. The motif was identified by MEME. **c** Metaplot profiles of normalized R330fs mutant ChIP-seq signals. The normalized signals in each GATA3-peak group were plotted. **d** Venn diagram showing the overlap between wild-type GATA3 peaks in CR3 (red) and R330fs mutant peaks (yellow). **e** Read density heatmap showing the ChIP-seq signals of R330fs mutant and wild-type GATA3 in each peak category (defined by peak overlap analysis shown in **d**). **f** Consensus and non-consensus GATA3-binding motifs. The box plots showing the number of motifs/peak in each peak group (R330fs-WT common, R330fs unique, WT unique). The bar graphs showing the frequency of peaks that contain the AGATBD motif or WGATAR motif. **g** Immunostaining of F-actin by phalloidin as observed by confocal laser scanning microscopy. Green: F-actin; blue: DAPI. **h** Quantitative analysis of actin filament formation shown in **g**. The length of actin fibers was measured by ImageJ software. Mean and SD are indicated in the scatter dot plot

and PR-B (Supplementary Fig. 6d), and tested the effect of progesterone. Exogenous expression of either PR-A or PR-B rescued progesterone sensitivity (Fig. 6e). Conversely, PR knockdown in control T47D cells generated a progesterone tolerance similar to GATA3-mutant cells (Fig. 6f; Supplementary Fig. 6e).

We extended these observations to a different system by altering expression levels of wild-type and mutant GATA3 in KPL-1 cells, which has a ZnFn2 mutation (D336fs). Depletion of endogenous GATA3 in these cells (both wild-type and mutant versions) partially rescued PR expression. Overexpression of wild-type

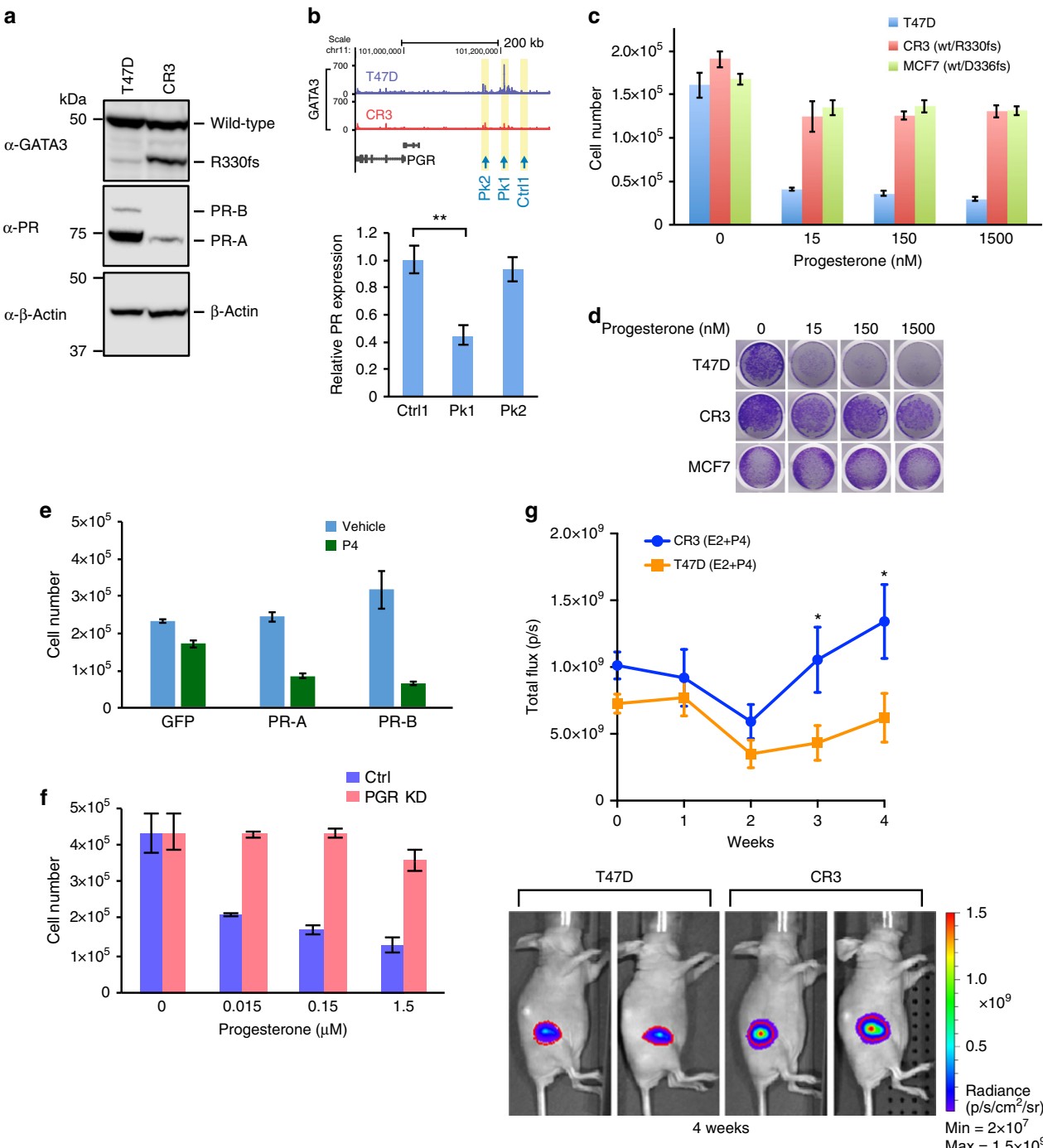

**Fig. 6** ZnFn2 mutant cells exhibit low PR expression, and growth advantage in the presence of progesterone. **a** Immunoblots showing the expression level of GATA3 (top), PR-B, and PR-A (middle, Cell Signaling). β-actin levels (bottom) are shown as a loading control. **b** PR gene expression analysis in the GATA3-peak depleted cells by CRISPR-Cas9. Relative expression was calculated by the delta–delta Ct method. \*\*$p < 0.01$, unpaired $t$-test. **c** Cell proliferation analysis of GATA3-mutant cells in the presence of progesterone. Data are presented as mean with SD. **d** Representative images of cell proliferation analysis. The cells are stained with crystal violet after the treatment with indicated concentration of progesterone. **e** Cell proliferation analysis of CR3 cells transduced with PR-A and PR-B expression vectors. **f** Cell proliferation analysis of PR knockdown cells in the presence of progesterone. **g** Tumor growth analysis by mouse xenograft experiments. Estrogen and progesterone pellets were implanted subcutaneously 1 week prior to cell injection, which was defined as time zero. Data are shown as mean with SEM ($N > 10$). \*$p < 0.05$, unpaired $t$-test

GATA3 further restored the expression of PR at both RNA and protein levels (Supplementary Fig. 6f–g). These data indicated that ZnFn2 mutations modulate PR expression and progesterone sensitivity in multiple cellular settings.

To further explore this apparent growth advantage in the presence of progesterone, we turned to the xenograft model. In this assay, progesterone pellets were simultaneously implanted into mice along with estrogen pellets. The GATA3-mutant

tumors exhibited a distinct growth advantage in vivo over wild-type T47D cells in the presence of progesterone pellets (Fig. 6g). Analysis of RNA extracted from the developing tumors at necropsy confirmed a reduced expression of PR in the R330fs mutant tumors (Supplementary Fig. 6h). These results suggest that PR is a major downstream target of GATA3, and its expression is downregulated in the presence of ZnFn2 mutations.

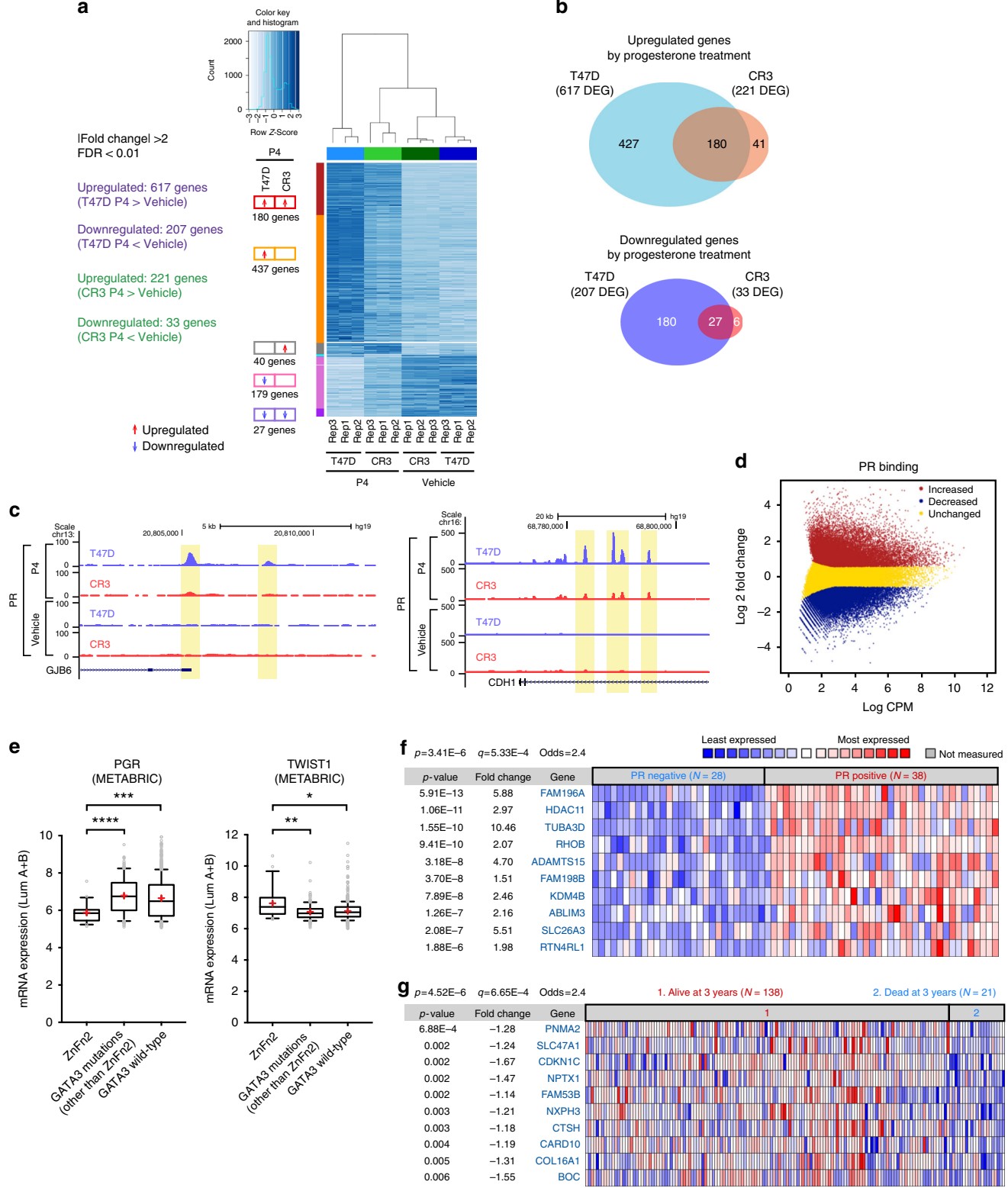

**An aberrant PR-regulatory network in GATA3-mutant tumors**. We used a systems approach to explore the molecular mechanisms by which decreased PR expression contributes to the phenotypic outcomes in the ZnFn2 mutants. RNA-seq revealed 824 genes (617 upregulated genes, 207 downregulated genes) as downstream targets of PR (Fig. 7a; Supplementary Data 1). The gene ontology analysis indicated that upregulated genes were significantly enriched with cellular movement and cellular development-related genes, including KRT4 and GJB2 (Supplementary Fig. 7a,b; Supplementary Data 1). Downregulated genes are significantly associated with tissue morphology, cellular homeostasis, and angiogenesis related pathways, including TWIST1 (Supplementary Fig. 7c; Supplementary Data 1). At an FDR < 0.01, |fold change| > 2, only 221 genes were upregulated, and 33 genes were downregulated in the mutant cells by progesterone treatment (Fig. 7a,b). Upon progesterone treatment, 1213 genes (561 genes upregulated, 652 genes downregulated) were differentially expressed between mutant and control T47D cells. We conducted RNA-seq analysis of xenograft tumors xenografts grown in the presence of progesterone pellets, and analyzed DEGs. The expression patterns within DEGs were largely conserved in tumors from the xenografts (Supplementary Fig. 7d–e).

To understand the contribution of PR action to gene expression responses, we determined the chromatin localization of PR in T47D control cells as well as GATA3-mutant cells. A large number (60,966) of PR-binding peaks were observed in control T47D cells after progesterone treatment (Fig. 7c). Notably, 39,639 PR-binding peaks were also detected in the mutant cells, and many of them were overlapped with PR peaks in T47D control cells. At a FDR < 0.05, |fold change| > 1.5, 18,175 loci were defined as part of an increased-binding group, and 16,214 loci were defined as the decreased-binding group, while 33,834 sites were unchanged (Fig. 7d). The GSEA analysis indicated that those loci displaying differential-binding levels were strongly correlated with differential gene expression between control and mutant T47D cells (Supplementary Fig. 7f).

Finally, to assess the relevance of our findings in these model systems, we returned to patient data. In both the METABRIC[4] and TCGA breast cancer cohorts[21], PR expression in ZnFn2 tumors was significantly lower than that in other GATA3-mutant tumors or in luminal breast tumors that express wild-type GATA3 (Fig. 7e; Supplementary Fig. 7g). Immunohistochemistry data in the TCGA cohort also supported lower expression of PR in ZnFn2 tumors (Supplementary Fig. 7h). TWIST1, which is upregulated by R330fs GATA3, was upregulated in ZnFn2 mutant tumors in a similar comparison (Fig. 7e; Supplementary Fig. 7g). GATA3 expression levels were similar in all tumors bearing GATA3 mutations, including ZnFn2 mutations (Supplementary Fig. 7g).

To elaborate on the gene expression analysis, we performed ONCOMINE Concept Analyses. As an input gene set, we used 437 genes uniquely upregulated by progesterone in control T47D cells but not in CR3 cells (Fig. 7a,b). Consistent with our in vitro data, expression of many of these genes were positively correlated with PR expression status in breast tumors. In the "Over-expression in Breast Carcinoma - Progesterone Receptor Positive[37]" data cohort, the expression data of 341 genes (out of 437 input genes) were available, and 94 genes were significantly ($p < 0.05$, Student's $t$-test) associated with PR-positive status (Fig. 7f). Gene ontology analysis also revealed a significant overlap between downregulation of those genes and aggressive tumor phenotypes, including worse prognosis and metastasis (Supplementary Table 3). In the case of the "Under-expression in Breast Carcinoma - Dead at 3 Years[38]" data, 53 genes out of 345 genes were downregulated in the category "patient deceased at 3 years" (Supplementary Data 1), and 38 genes were categorized in the top 5% under-expressed genes (Fig. 7g; Supplementary Table 3).

## Discussion

Recent large-scale breast cancer genomic profiling identified frequent mutations in the GATA3 gene. However, the clinical and molecular outcomes of GATA3 mutations are poorly understood, because there are very few studies describing the functions of mutant GATA3[23,24,39,40]. In this study, classification of the GATA3 mutations revealed different molecular and clinical outcomes. Overall, patients with tumors with GATA3 mutations live longer than patients with wild-type GATA3. GATA3 mutations found in zinc-finger 2, conversely, lead to distinct features: frequent detection in luminal B tumors and worse prognosis. Splice site and truncation mutations (found in Exons 5–6) are frequently detected in luminal A; these patients have better prognosis than those with wild-type GATA3. The differential outcomes between splice site mutations and ZnFn2 mutations suggest that the impact of complete loss of the second zinc-finger motif differs from that of partial loss.

We utilized the CRISPR–Cas9 system to investigate the molecular outcomes induced by ZnFn2 mutations. The GATA3-mutant cells became more aggressive and exhibited faster tumor growth in vivo. Transcriptomic analysis identified altered transcriptional programs in the mutant cells, marked by downregulation of PR and upregulation of epithelial to mesenchymal regulators such as TWIST1 and SLUG. Previous studies suggested that the mutations in the second zinc-finger domain are loss-of-function mutations, as they are located within the critical DNA-binding domain and impact DNA-binding activity[22,39,40]. Indeed, decreased GATA3 binding was observed, particularly in loci where the chromatin accessibility was lower at baseline. At other loci, the GATA3-mutant-induced binding gains that were significantly correlated with increased gene expression levels. Interestingly, R330fs mutant ChIP-seq signals were enriched in a

**Fig. 7** Progesterone regulatory network is impaired in R330fs mutant cells. **a** Clustered heatmap showing the differential gene expression (DEG) by progesterone (150 nM, 3 h) treatment in T47D and CR3 cells. The genes are first grouped by DEG status (upregulation and downregulation are indicated as red and blue arrows respectively), then sorted by decreasing FPKM within each group. **b** Upregulated and downregulated genes in T47D cells and CR3 cells after progesterone treatment. Venn diagram represent the overlap between differentially expressed genes in T47D cells and CR3 cells. **c** Representative genome browser tracks at differential PR-binding sites. UCSC Genome Browser views showing the mapped read coverage of PR ChIP-seq. **d** Differential-binding analysis of PR-binding events in control T47D versus CR3 cells. Increased, decreased, and unchanged binding events are indicated in red, blue, and yellow respectively. CPM counts per million. **e** Box-and-whisker plots displaying the PR and TWIST1 expression levels in GATA3-mutant tumors. Expression data were collected from the luminal A and luminal B tumors in each GATA3 status category (the METABRIC cohort). Whiskers indicate 10th and 90th percentiles. Means and medians are shown as plus signs and black lines, respectively. *$p < 0.05$, **$p < 0.01$, ***$p < 0.001$, ****$p < 0.0001$, Mann–Whitney test. **f–g** The oncomine concept analyses. 437 uniquely upregulated genes in progesterone-treated T47D cells were used as the input gene set. $q$-value: estimated false discovery rates. **f** Heatmap showing the positive correlation between the uniquely upregulated genes and PR status in the clinical gene expression data. The top 10 genes (ranked by $p$-values) are shown. **g** Heatmap showing the underexpression of the uniquely upregulated genes in the aggressive tumors (patients died at 3 years). The top 10 genes are shown

novel-binding motif (AGATBD), and higher in the increased total GATA3-binding group as compared to the other groups. The R330fs mutant still contains an intact N-terminal zinc-finger domain, which is known to have higher affinity to the "AGATB" DNA sequence[41]. Therefore, the mutant may utilize the N-terminal zinc-figure domain for its chromatin localization. Because R330fs mutant was localized at both increased and decreased GATA3-binding sites, R330fs mutant may influence chromatin-binding activity of wild-type GATA3.

Our in vitro and in vivo model system identified progesterone receptor as a key downstream target of GATA3. Gene expression and chromatin profiling indicated that PR regulates expression levels of epithelial and mesenchymal marker genes. The ZnFn2 mutant decreases PR expression and disorganizes PR action on chromatin. Consistently, the clinical gene expression data displayed lower expression of PR in ZnFn2 mutant tumors, compared to the other luminal tumors. Many PR-regulatory genes defined from our system had significant correlation with PR status in patients, and downregulation of a subset of those genes are associated with worse prognosis. These findings highlight the role of PR as critical to the maintenance of epithelial identity in luminal breast cancer.

## Methods

**Cell line and cell culture**. The T47D and MCF7 cells were obtained from ATCC. The KPL-1 cell line was kindly provided by Dr. John Colicelli. All cell lines were cultured in DMEM high-glucose medium with 10% FBS (Thermo Fisher Scientific) at 37 °C (with 10% CO₂). For the xenograft experiments, the Luc2-tdTomato fusion gene was transduced into T47D cells using the pHAGE lentiviral vector. The Luc2-tdTomato fusion gene was obtained from the pcDNA3.1(+)/Luc2=tdT vector (a gift from Dr. Christopher Contag) (Addgene plasmid #32904)[43]. The pHAGE vector, PR shRNA, and GATA3 shRNA vectors were kindly provided by Dr. Guang Hu (NIEHS/NIH). pRR-PR-5Z was a gift from Charles Miller (Addgene plasmid # 23057)[44], and used to generate PR-A and PR-B expression vector.

**Generation of GATA3-mutant cell line**. The guide RNA was designed by the Dr. Feng Zhang Laboratory webtool (crispr.mit.edu). The double-stranded oligonucleotides were inserted into the PX458 plasmid. pSpCas9(BB)-2A-GFP (PX458) was a gift from Dr. Feng Zhang (Addgene plasmid # 48138)[45]. The homology donor dsDNA was generated by the SLIC cloning method[46]. The primers were listed in Supplementary Table 4. T47D cells were co-transfected with the PX458 vector and amplified donor DNAs. Wild-type or R330fs mutation (2-nucleotide deletions at Chromosome 10: 8,111,563-8,111,565) containing donor DNA was used for the generation of CRctrl or CR3 respectively. The antibiotics selection was started 72 h after transfection. After a 1 week treatment with blasticidin, single-cell clones were isolated, and positive clones were screened by PCR. The antibiotics gene cassette was removed by FLP recombination using the pSICO-Flpo vector (a gift from Dr. Tyler Jacks laboratory, Addgene plasmid # 24969)[47].

**Xenograft studies**. All animal studies were approved by the NIEHS Animal Care and Use Committee and conducted in accordance with The Guide to the Care and Use of Laboratory Animals, 8th edition. All experiments were blinded. Four-week-old BALB/c nude female mice were obtained from Charles River and maintained on an alfalfa-free diet in the NIEHS animal facility. 17β-estradiol (90 day release, 0.72 mg) pellets and/or progesterone (90 day release, 10 mg per pellet) pellets (Innovative Research of America) were implanted subcutaneously 1 week before cell injection. One million cells (expressing the Luc2-tdTomato fusion protein) were resuspended in PBS(−) and mixed in a 1:1 ratio with growth factor reduced, LDEV-free matrigel (Corning). The cell suspension (100 μL/animal) was injected into inguinal mammary fat pads (N = 15). Tumor growth was monitored and measured once a week by bioluminescence signals 20 min after intraperitoneal luciferin injection (150 mg/kg body weight) using the Spectrum In Vivo Imaging System (PerkinElmer). The bioluminescence signals were quantified by Living Image version 4.4 software (PerkinElmer). Regions of interest (ROI) were determined by the Auto ROI tool (threshold parameter: 3%), and quantified as photons/second.

**Enhancer deletion**. The guide RNAs were designed by the Feng Zhang Laboratory web tool. Annealed oligonucleotides were ligated into the lentiGuide-Puro vector (a kind gift of Dr. Feng Zhang (Addgene plasmid # 52963)[48]. The viruses were produced by transient transfection of 293 T cells with the Cas9 vector, psPAX2, and pMD2.G plasmids. psPAX2 and pMD2.G were gifts from Dr. Didier Trono (Addgene plasmid #12260, #12259). To enhance the target deletion efficiency, T47D cells were co-infected with two viruses carrying two different gRNAs, which

target 200 to 300 bp flanking regions (Supplementary Table 4). The stable cell pools were established by a 2-week selection with Puromycin.

**ChIP-seq**. Cells were fixed with 1% formaldehyde at room temperature for 10 min and quenched with glycine. Fixed cells were further treated with hypotonic buffer containing 10 mM HEPES-NaOH pH 7.9, 10 mM KCl, 1.5 mM MgCl₂, 340 mM sucrose, 10% glycerol, 0.5% Triton X-100, and Halt Protease & Phosphatase Single-Use Inhibitor Cocktail (Thermo Fisher Scientific), and resuspended in lysis buffer (20 mM Tris-HCl pH 8.0, 2 mM EDTA, 0.5 mM EGTA, 0.5 mM PMSF, 5 mM sodium butyrate, 0.1% SDS, and protease inhibitor cocktail). Chromatin was fragmented by sonication with Covaris S220, then diluted to adjust the SDS concentration to 0.05%. Immunoprecipitation was performed with indicated antibodies (Supplementary Table 5). A protein A/G Dynabeads mixture (1:1 ratio) was used to capture antibodies. Eluted DNA was reverse crosslinked at 65 °C, followed by the incubation with proteinase K. DNA was purified by AMPure XP (Beckman Coulter). The sequencing libraries were prepared by the NEXTflex Rapid DNA-seq kit (Bioo Scientific Corporation) and sequenced on NextSeq 500 (Illumina) at the NIEHS Epigenomics Core Facility. Reads were filtered based on a mean base quality score >20, and mapped to hg19 genome by Bowtie 0.12.8 allowing only uniquely mapped hits[49]. Duplicate reads were removed using MarkDuplicates.jar from picard-tools-1.107 package. Paired-end reads were converted to a single fragment for metaplot analyses and visualization on the UCSC Genome Browser. For the PR ChIP-seq, the cells were treated with 150 nM progesterone or the same volume of the vehicle (ethanol) for 3 h.

**Peak call**. Peak calling analyses were performed using HOMER v4.1 with default parameters[50]. Only the first read of each read pair was used for peak calling.

**Pathway analysis**. All (Fig. 7) or the top 100 (Supplementary Fig. 2) DEGs were used to perform the IPA (Ingenuity Pathway Analysis) gene ontology analyses (QIAGEN). For the identification of the association with cancer related gene expression patterns indicated in Supplementary Table 1, 2, all DEGs (|fold change| > 2, FDR < 0.01) were applied to the ONCOMINE Concept Analyses (Thermo Fisher Scientific), while 437 uniquely upregulated genes in progesterone-treated T47D cells were used for the analysis in Fig. 7f–g and Supplementary Table 3.

**Peak classification**. EdgeR[51] was used to classify the ChIP-seq peaks into three groups (Increased, Decreased, Unchanged) based on the ChIP-seq signal differences between control cells and CR3 cells. Read counts per 400 bp window, centered on peaks, were collected then evaluated with the GLM method. Classification thresholds were set at FDR < 0.05 and |fold change| > 1.5 For heatmap visualization, read counts were normalized to 35 million total mapped fragments per sample.

**Clinical data analysis**. The METABRIC clinical data and gene expression/mutation data were obtained from the Cancer Genomics Data Server R (cgdsr) package (hosted by the Computational Biology Center at Memorial-Sloan-Kettering Cancer Center, the data were obtained on October 24th 2016) and cBio Cancer Genomics Portal (http://cbioportal.org). For the PAM50 subtype analyses of the METABRIC cases, we used the previously defined classification, described in[52]. The TCGA breast cancer data were obtained from the public TCGA data portal.

**Immunostaining**. For the F-actin staining, cells were fixed with 2% formaldehyde for 15 min at room temperature and permeabilized with PBS buffer containing 0.1% Triton-X100. The cells were further incubated with Phalloidin CruzFluor 488 Conjugate (Santa Cruz Biotechnology) and mounted in Vectashield mounting medium with DAPI.

**Immunoblotting**. Cells were lysed with sample loading buffer, and extracted proteins were separated by SDS-PAGE. Anti-GATA3 (Cell Signaling), anti-progesterone receptor (Cell Signaling), and anti-b-actin (Abcam) antibodies were used as primary antibodies. Anti-rabbit IgG or mouse IgG HRP-conjugated antibody (GE Healthcare) was used as a secondary antibody. The chemiluminescent signals were detected by the Odyssey Fc Imaging System, and analyzed by the Image Studio software (LI-COR).

**Cell proliferation assay**. Cells were resuspended in 5% FBS-containing DMEM and spread on a 24-well plate (50,000 cells/well). After incubation at 37 °C for 24 h, the medium was replaced to progesterone-containing or vehicle(ethanol)-containing medium. The medium was replaced every day for 3 days, at which time the total cell number was counted using a Coulter counter (Beckman).

**Gene expression analysis**. Total RNAs from cultured breast cancer cells were purified by RNeasy Kit or miRNeasy kit (QIAGEN). For the progesterone stimulation, cells were incubated with 150 nM progesterone (or ethanol as a control) for 3 h. Total RNAs from tumor tissues were extracted by QIAzol Reagent and purified by miRNAeasy kit (QIAGEN). The purified RNA concentration was measured by a

NanoDrop 1000 Spectrophotometer (Thermo Fisher Scientific). The cDNAs were synthesized by using iScript cDNA Synthesis Kit (Bio-Rad Laboratories). Quantitative PCR was performed using the CFX384 Real-Time PCR Detection System (Bio-Rad Laboratories). Relative expression was calculated by delta–delta Ct method. TBP was used as a reference gene for normalization.

**RNA-seq**. Library preparation and sequencing were performed by Q2 Solutions on an Illumina platform (HiSeq 2000). Libraries for RNA-seq analysis of tumors excised from the mouse xenograft model were prepared by TruSeq RNA Library Prep Kit (Illumina), and sequenced on NextSeq 500 (Illumina) at the NIEHS Epigenomics Core Facility. Raw read pairs were filtered to require a minimum average base quality score of 20. Filtered read pairs were mapped against the hg19 reference genome by STAR (version 2.5) with the following parameters: -- out-SAMattrIHstart 0 --outFilterType BySJout --alignSJoverhangMin 8 -- out-MultimapperOrder Random (other parameters at default settings)[53]. Mapped read counts per gene were collected by Subread featureCounts (version 1.5.0-0-p1). DEGs were identified with DESeq2 (v1.10.1) using filtering thresholds of FDR < 0.01 and |fold change| > 2[54]. The gene models are taken from hg19 RefSeq annotations, downloaded from the UCSC Genome Browser on 28 March 2016. DESeq2 (v1.10.1) was also used to generate a clustered heatmap for visualizing sample similarity, where sample-to-sample distances were calculated for normalized counts from variance stabilizing transformation. Similarity is scored as the Euclidean distance between sample pairs. Thus, sample pairs with similar counts per gene across all gene models will have a low distance score (corresponding to dark blue color).

For the GSEA analysis, we filtered the expression data to only include genes with a nonzero read count for all samples. We then performed a GSEA analysis, using the GSEA Java software version 2.1.0 and default parameters, with this expression data set and various gene lists obtained from ChIP-seq peaks. We associated the ChIP-seq peaks with genes by identifying the closest RefSeq TSS to each peak center. We excluded any peaks that were more than 50 kbp from the nearest TSS.

**ATAC-seq**. 25,000 cells were incubated in CSK buffer (10 mM PIPES pH 6.8, 100 mM NaCl, 300 mM sucrose, 3 mM MgCl$_2$, 0.1% Triton X-100) on ice for 5 min. An aliquot of 2.5 µl of Tn5 Transposase was added to a total 25 µl reaction mixture. After PCR amplification (eight total cycles), DNA fragments were purified with AMPure XP (1:3 ratio of sample to beads). The libraries were sequenced on NextSeq 500 at the NIEHS Epigenomics Core Facility. Reads were filtered based on a mean base quality score > 20. After adapter trimming by Trim Galore! (Babraham Institute), reads were mapped to hg19 genome using Bowtie 0.12.8[55]. Unique reads (non-duplicate reads) were used for the subsequent analysis.

**Simulating the solvated mutant structure**. X-ray crystal structure of DNA-bound human GATA transcription factor complex (PDB: 4HCA) was used as a template to model the mutant structure (with the sequence   MGRECVNC-GATSTPLWRRDGTGHYLCNACGLYHKMNGQNRPLIKPKRR LSAARRA GTSCANCQTTTTTLWEECQWGPCLQCLWALLQASQY). Missing and non-identical residues were introduced using the program, Modeller V9.14[56]. The resultant mutant structure was first energy minimized in vacuum and then subjected to a lengthy molecular dynamics simulation using the program, Amber.14[57]. Prior to energy minimization, missing protons were added and then, the mutant was energy minimized using Amber force field, FF14SB[57]. The optimized structure was then solvated in a box of water (with 8357 molecules). Prior to equilibration, the system was subjected to: (1) a nanosecond of constrained molecular dynamics with the backbone of the mutant peptide constrained to the original positions with a force constant of 10 kcal/mol/nm, (2) minimization, (3) low temperature constant pressure simulation to assure a reasonable starting density, (4) minimization, (5) step-wise heating at constant volume, and (6) a 2 ns constant volume, constant temperature molecular dynamics run. The final structure was then subjected to a 10 ns constant temperature simulation to obtain a reasonable solvated structure for the mutant.

**Quantification and statistical analysis**. Kaplan–Meier survival curves, box plots were generated by Prism 6 (GraphPad Software). p-values were calculated by Prism 6 (GraphPad Software). Unpaired t-tests were applied to Figs. 2e, 5h and 6g, Supplementary Figures 6f and 6g. Log rank tests were applied to Fig. 1c and Supplementary Figure 1g. Fisher's exact tests were applied to Supplementary Figures 1e and 7h. Mann–Whitney tests were applied to Figs. 1f, 5f and 7e and Supplementary Fig. 7g. $\chi^2$ tests were applied to Supplementary Fig. 4e.

**Data availability**. Data pertinent to this study are available at Gene Expression Omnibus under Accession Number GSE99479. (https://www.ncbi.nlm.nih.gov/geo/query/acc.cgi?acc=GSE99479). Uncropped immunoblot images are shown in Supplementary Figure 8.

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

## Acknowledgements

We gratefully acknowledge NIEHS/NIH core groups (Flow Cytometry Center, Pathology Support Group, Comparative Medicine Branch, Epigenomics core, Fluorescence Microscopy and Imaging Center, Viral Vector Core, Integrated Bioinformatics core) for outstanding technical assistance. We thank Mr. Chris McGee for help with animal in vivo imaging; Dr. Kyathanahalli Janardhan and Ms. Natasha P. Clayton, for advice and analysis of animal tissues; Dr. Grace Kissling for advice on statistical analysis. We are indebted to Dr. Takashi Shimbo (Osaka University) for technical advice. This work was supported, in part, by the Intramural Research Program of the National Institute of Environmental Health Sciences, NIH (ES101965 to PAW, ES043010 to L.P.), by the NCI Breast SPORE program (P50-CA58223-09A1 to C.M.P) and the Breast Cancer Research Foundation (C.M.P.).

## Author contribution

Experiments in this study were conceived by M.T., J.D.R. and P.A.W. Experiments were performed by M.T., J.D.R., K.C. and P.M. Data analysis was performed by M.T., S.A.G., C.J.T., C.M.P. and P.A.W. Molecular modeling was performed by L.P. All authors participated in composing and editing the manuscript. All authors read and approved the final manuscript.

## Additional information

**Competing interests:** The authors declare no competing financial interests.

