## [Peer Review File · Nature Communications]

Reviewers' comments:

Reviewer #1 (Remarks to the Author):

The authors have identified a critical role for GATA3 mutations in breast cancer and associated outcomes.

1. GATA3 mutations have differential functions and in particular, mutations in the a zinc finger region shift binding of GATA3 from canonical regulatory binding sites to new sites that results in downregulation of the progesterone receptor, a key regulator of epithelial/luminal cell fate.
2. The authors have composed a very thorough and convincing link between GATA3 mutations and epithelial to mesenchymal gene expression profiles.
3. In addition, the investigators have shown that GATA3 expression is significantly associated with better outcome in luminal breast cancer and presence of certain mutations (ZnFn2) cause shifts from a more epithelial-like cancer (luminal A) to a more aggressive and difficult to treat cancer (luminal B). These findings delineate a critical role for GATA3 mutations in breast cancer heterogeneity and may provide a causal role for why some breast cancers transition from epithelial to mesenchymal phenotypes.
4. The manuscript provided thorough statistical analysis and used at least models of luminal breast cancer cell lines.
5. The use of global expression profiling and ChIP-sequencing increased the scientific rigor and provided confidence that the investigators used non-biased approaches to analyze the datasets and test a hypothesis.
6. The main concern which dampens enthusiasm is the lack of clear biological evidence that different GATA3 mutations cause clear shifts from luminal to mesenchymal cell fates. Phenotypic results besides just tumor growth would provide strong supporting evidence for the findings. Expression of cytokeratins, Epcam, CD44, CD24 to name a few would increase confidence.

Reviewer #2 (Remarks to the Author):

This manuscript of Takaku et al studies GATA3, a pioneer factor found mutated in ~10% of breast cancers. GATA3 promotes gene transcription programs related to mesenchymal-to-epithelial transition and its mutations were previously thought to be generally loss of function ones. Authors now found a set of mutations at the 2nd zinc finger of GATA3 (namely, ZnFn2 mutation or R330fs as the most prevalent form) behave differently in multiple cohorts of human cancer patients and have gain of function activity in several cancer cell model systems, including T47D-derived CR3 and ExoR330fs cells that harbor CRISPR-Cas9 mediated ZnFn2 mutation and R330fs overexpression, respectively. In particular, GATA3 ZnFn2 mutation correlates with worse prognosis. Heterozygous GATA3 ZnFn2/R330fs mutation introduced into CR3/T47D cells resulted in profound changes in cell morphology/malignant growth, transcriptome (notably PR down-regulation and TWIST/Slug upregulation), GATA3 chromatin binding relative to WT, chromatin accessibility, and response to P4. Mechanistically, authors used isoform specific versus common antibodies of GATA3 proteins to show that those with ZnFn2/R330fs mutation show distinctive occupancy patterns, with the increased binding to a unique motif (AGATBD), which associates with the up-regulated genes. Lastly, authors carried out correlation analyses to show the dampened PR response is crucial for cancer-promoting functions associated with ZnFn2/R330fs mutation using both T47D cell models and human patient cohorts.

In summary, this work really is a tour de force study, using an integrated approach that includes CRISPR-Cas9 technology based editing of GATA3 and PR enhancers, in vitro/vivo cancer models, the comprehensive genomics profilings (ChIP-seq, ATAC-seq and RNA-seq) and human patient dataset mining to delineate mechanistic details of a crucial GATA3 mutation and the underlying oncogenic pathways. Report of the gain-of-function of GATA3 ZnFn2/R330fs mutation is both novel and significant to the breast cancer field. Data presented here are well described with convincing statistics. The implication of the work is far reaching in terms of therapeutics because ZnFn2

mutation can be potentially targeted. Therefore, the report shall appeal to the field and readers. In principle it is suitable for publication at Nature Communications, after addressing the below comments.

Comments:

1/ Abstract: it is better to specify the GATA3 ZnFn2 mutation as the main focus of the study, if the word limit is not an issue here.

2/ Supplemental Fig. 2c: author needs to explain how the genotyping of GATA3 wt vs mutant was done since T47D_R330fs is heterozygous in cells.

3/ Figure reference: authors need to pay attention and refer to the correct figure panels. For example, starting from line 210, fig 3a-c shall be fig 4a-c, and so on at multiple places.

4/ Fig 4-5: differential binding of WT vs R330fs mutation GATA3 needs to be validated by ChIP-qPCR of at least 1 unique locus for each, such as PR and HDAC9.

5/ Fig 4d: did authors identify enrichment of the motif AGATBD among increased peaks, relative to decreased peaks? Related to this and in Fig 5c, it would be more direct to generate a Venn diagram using the 6,239 R330fs ChIPseq peaks and the increased peaks shown in Fig 4c (5,161 peaks). Essentially what is the relative percentage of WT vs R330fs binding among the increased and decreased peaks shown in Fig 4c.

6/ Fig 7: does PR binding show correlation to GATA3 binding?

7/ Discussion section: Authors want to discuss over the mainly heterozygous nature of GATA3 ZnFn2 mutation. For example CR3 cells have a WT and a mutant copy of allele both producing proteins. Does ZnFn2 mutation affect activity/binding of WT GATA3 (related to the above comment 5)?

Minor issues:

1/ line 259: "under physiological control" is a typo?

The point-by-point responses to the comments are as follows.
We thank both reviewers for their insightful comments. These reviews have substantially improved our manuscript quality.

Reviewers' comments:

Reviewer #1 (Remarks to the Author):

The authors have identified a critical role for GATA3 mutations in breast cancer and associated outcomes.

1. GATA3 mutations have differential functions and in particular, mutations in the a zinc finger region shift binding of GATA3 from canonical regulatory binding sites to new sites that results in downregulation of the progesterone receptor, a key regulator of epithelial/luminal cell fate.

2. The authors have composed a very thorough and convincing link between GATA3 mutations and epithelial to mesenchymal gene expression profiles.

3. In addition, the investigators have shown that GATA3 expression is significantly associated with better outcome in luminal breast cancer and presence of certain mutations (ZnFn2) cause shifts from a more epithelial-like cancer (luminal A) to a more aggressive and difficult to treat cancer (luminal B). These findings delineate a critical role for GATA3 mutations in breast cancer heterogeneity and may provide a causal role for why some breast cancers transition from epithelial to mesenchymal phenotypes.

4. The manuscript provided thorough statistical analysis and used at least models of luminal breast cancer cell lines.

5. The use of global expression profiling and ChIP-sequencing increased the scientific rigor and provided confidence that the investigators used non-biased approaches to analyze the datasets and test a hypothesis.

6. The main concern which dampens enthusiasm is the lack of clear biological evidence that different GATA3 mutations cause clear shifts from luminal to mesenchymal cell fates. Phenotypic results besides just tumor growth would provide strong supporting evidence for the findings. Expression of cytokeratins, Epcam, CD44, CD24 to name a few would increase confidence.

We appreciate the comments from this reviewer and we agree that adding emphasis on EMT would strengthen the findings. We have now added heatmaps in Supplementary figure 2e showing gene expression changes of representative mesenchymal marker genes as well as luminal and epithelial marker genes in GATA3 mutant cells. We interpret these data as indicative of a shift towards mesenchymal cell fate in GATA3 mutant (CR3) cells.

Reviewer #2 (Remarks to the Author):

This manuscript of Takaku et al studies GATA3, a pioneer factor found mutated in ~10% of breast cancers. GATA3 promotes gene transcription programs related to mesenchymal-to-epithelial transition and its mutations were previously thought to be generally loss of function ones. Authors now found a set of mutations at the 2nd zinc finger of GATA3 (namely, ZnFn2 mutation or R330fs as the most prevalent form) behave differently in multiple cohorts of human cancer patients and have gain of function activity in several cancer cell model systems, including T47D-derived CR3 and ExoR330fs cells that harbor CRISPR-Cas9 mediated ZnFn2 mutation and R330fs overexpression, respectively. In particular, GATA3 ZnFn2 mutation correlates with worse prognosis. Heterozygous GATA3 ZnFn2/R330fs mutation introduced into CR3/T47D cells resulted in profound changes in cell morphology/malignant growth, transcriptome (notably PR down-regulation and TWIST/Slug upregulation), GATA3 chromatin binding relative to WT, chromatin accessibility, and response to P4. Mechanistically, authors used isoform specific versus common antibodies of GATA3 proteins to show that those with ZnFn2/R330fs mutation show distinctive occupancy patterns, with the increased binding to a unique motif (AGATBD), which associates with the up-regulated genes. Lastly, authors carried out correlation analyses to show the dampened PR response is crucial for cancer-promoting functions associated with ZnFn2/R330fs mutation using both T47D cell models and human patient cohorts.

In summary, this work really is a tour de force study, using an integrated approach that includes CRISPR-Cas9 technology based editing of GATA3 and PR enhancers, in vitro/vivo cancer models, the comprehensive genomics profilings (ChIP-seq, ATAC-seq and RNA-seq) and human patient dataset mining to delineate mechanistic details of a crucial GATA3 mutation and the underlying oncogenic pathways. Report of the gain-of-function of GATA3 ZnFn2/R330fs mutation is both novel and significant to the breast cancer field. Data presented here are well described with convincing statistics. The implication of the work is far reaching in terms of therapeutics because ZnFn2 mutation can be potentially targeted. Therefore, the report shall appeal to the field and readers. In principle it is suitable for publication at Nature Communications, after addressing the below comments.

Comments:

1/ Abstract: it is better to specify the GATA3 ZnFn2 mutation as the main focus of the study, if the word limit is not an issue here.

Thank you very much for this comment. We modified the abstract and specified ZnFn2 mutation.

2/ Supplemental Fig. 2c: author needs to explain how the genotyping of GATA3 wt vs mutant was done since T47D_R330fs is heterozygous in cells.

We apologize for insufficient explanation. We first amplified the GATA3 Exon 5 region from the CR3 clone, and the PCR product was cloned into a vector. The mutation was analyzed by sanger sequencing. 3 out of 10 clones contain 'GA' nucleotide deletion (Supplementary fig. 2c), while other clones contain the wild-type sequence. We updated figure legend to clarify the method.

3/ Figure reference: authors need to pay attention and refer to the correct figure panels. For

example, starting from line 210, fig 3a-c shall be fig 4a-c, and so on at multiple places.

We apologize for these errors. We corrected the manuscript.

4/ Fig 4-5: differential binding of WT vs R330fs mutation GATA3 needs to be validated by ChIP-qPCR of at least 1 unique locus for each, such as PR and HDAC9.

We appreciate this suggestion. We conducted ChIP-qPCR at representative loci (PR, HDAC9, and BMPER), and confirmed differential GATA3 binding at those loci (Supplementary fig. 4b).

5/ Fig 4d: did authors identify enrichment of the motif AGATBD among increased peaks, relative to decreased peaks? Related to this and in Fig 5c, it would be more direct to generate a Venn diagram using the 6,239 R330fs ChIPseq peaks and the increased peaks shown in Fig 4c (5,161 peaks). Essentially what is the relative percentage of WT vs R330fs binding among the increased and decreased peaks shown in Fig 4c.

Thank you very much for this insightful suggestion. We calculated AGATBD motif frequency and number, and both frequency and number were slightly higher at decreased binding peaks as compared to increased binding peaks.

Among 6239 R330fs peaks, 734 peaks were overlapped with increased GATA3 peaks, while 570 peaks were overlapped with decreased GATA3 peaks. Thus, the overlap ratio was slightly higher between R330fs peaks and increased GATA3 peaks. These results together with Fig.5c suggested that R330fs mutant contributes both increased and decreased GATA3 binding in CR3 cells.

Inner circle: distribution of total GATA3 peaks in each peak group
 Outer circle: distribution of R330fs peak (overlapped with GATA3 peaks) in each peak group

6/ Fig 7: does PR binding show correlation to GATA3 binding?

Thank you very much for the comment. To look at correlation of PR and GATA3 binding, we generated scatter plots at GATA3 binding sites and PR binding sites. In the absence of progesterone treatment, they showed moderate positive correlation. On the other hand, PR binding signals after P4 treatment didn't show strong correlation with GATA3 ChIP-seq signals.

We also looked at PR binding within differential GATA3 binding sites. In some cases (e.g. increased GATA3 peaks in P4-treated cells and decreased GATA3 peaks in vehicle-treated cells), PR binding showed mild correlation.

7/ Discussion section: Authors want to discuss over the mainly heterozygous nature of GATA3 ZnFn2 mutation. For example CR3 cells have a WT and a mutant copy of allele both producing proteins. Does ZnFn2 mutation affect activity/binding of WT GATA3 (related to the above comment 5)?

We appreciate this comment. Considering the above results (particularly in the comment 5 part), we modified abstract and discussion. We also added a discussion about a potential impact of the ZnFn2 mutant on wild-type GATA3 activity.

Minor issues:

1/ line 259: “under physiological control” is a typo?

We apologize for this typo. We corrected the text.

REVIEWERS' COMMENTS:

Reviewer #1 (Remarks to the Author):

The revised manuscript has addressed my main concern regarding evidence for EMT markers in cell lines expressing GATA3 mutations.

Reviewer #2 (Remarks to the Author):

Authors have adequately addressed the raised issues.